# MedAccessX: A Blockchain-Enabled Dynamic Access Control Framework for IoMT Networks

**DOI:** 10.3390/s25061857

**Published:** 2025-03-17

**Authors:** Guoyi Shi, Minfeng Qi, Qi Zhong, Ningran Li, Wanxin Gao, Lefeng Zhang, Longxiang Gao

**Affiliations:** 1Faculty of Data Science, City University of Macau, Macau 999078, China; d23091101564@cityu.edu.mo (G.S.); mfqi@cityu.edu.mo (M.Q.); wanxingao@cityu.edu.mo (W.G.); lfzhang@cityu.edu.mo (L.Z.); 2School of Computer and Mathematical Sciences, The University of Adelaide, Adelaide, SA 5005, Australia; ningran.li@adelaide.edu.au; 3Key Laboratory of Computing Power Network and Information Security, Qilu University of Technology, Jinan 250316, China; gaolx@sdas.org

**Keywords:** access control, blockchain, RBAC, ABAC

## Abstract

The integration of Internet of Things (IoT) devices in healthcare has enhanced medical efficiency but poses challenges such as data privacy risks and internal abuse. Traditional IoT data access frameworks suffer from centralization, limited scalability, and static permission controls. To address these issues, we propose MedAccessX, a blockchain-based access control framework combining attribute-based access control (ABAC) and role-based access control (RBAC). MedAccessX utilizes four types of smart contracts: a user management contract (UMC) for managing user operations, a medical data management contract (MDMC) for handling data, a policy contract (PC) for managing access rights, and an access control contract (ACC) for enforcing permissions and facilitating data sharing. Our evaluation, conducted on a private Ethereum blockchain network with multiple nodes, assesses security, deployment cost, gas consumption, throughput, and response time. Comparative analysis demonstrates that MedAccessX achieves lower deployment costs and higher throughput, outperforming existing solutions.

## 1. Introduction

The rapid proliferation of the Internet of Medical Things (IoMT) devices is revolutionizing the healthcare industry, with a growing number of sensors, wearables, and medical equipment continuously generating large volumes of health data. For example, smart insulin pumps, remote cardiac monitors, and wearable fitness trackers generate real-time health data that can provide critical insights into patient health. These devices continuously track vital signs, such as blood glucose levels and oxygen saturation, and transmit this critical information to healthcare providers for analysis and decision-making.

According to a recent report [1], the global IoMT market is expected to grow from $41 billion in 2020 to $158 billion by 2027, with over 30 billion connected medical devices predicted by 2025. The data generated by these devices is staggering: for example, a single hospital’s IoMT devices can generate terabytes of data daily [2]. This explosion of IoMT data holds significant potential to improve patient care, enhance diagnostics, and streamline medical processes. However, it also raises critical challenges, especially around data accessibility, security, and privacy. Sensitive patient information, such as real-time monitoring data or diagnostic reports, must be accessed by authorized personnel across multiple healthcare providers while maintaining strict privacy controls. Yet, in real-world applications, ensuring such controlled access is complicated by the fragmented nature of healthcare systems.

**Challenges of IoMT data sharing:** One prominent challenge arises from *the lack of seamless data sharing between hospitals (i.e., interoperability)*. Consider a common scenario where a patient, Alice, undergoes a medical examination at Hospital A but requires further treatment at Hospital B. If Hospital B cannot access Alice’s records from Hospital A, she may need to repeat tests, incurring unnecessary delays and expenses. This issue reveals a broader systemic challenge in the IoMT ecosystem: non-interoperable healthcare systems prevent the efficient exchange of medical data, leading to inefficiencies and hindering timely medical care.

In addition, *privacy and security* are important. IoMT devices collect vast amounts of sensitive personal information, and unauthorized access to such data could have devastating consequences for both patients and healthcare providers [3]. For instance, if a hacker gains access to a patient’s real-time monitoring device, such as a pacemaker, they could alter the device’s settings, leading to life-threatening situations. Unauthorized access to medical records could also result in identity theft, insurance fraud, or the exposure of highly sensitive personal information. In 2017, a ransomware attack [4] on the UK’s National Health Service (NHS) affected thousands of connected devices, compromising patient data and disrupting hospital services. It illustrates the serious risks associated with unsecured medical data in IoMT environments.

**Issues of current solutions:** Traditional access control methods, such as role-based access control (RBAC) [5] and discretionary access control (DAC) [6], were originally designed for static and isolated environments like corporate networks. While these models work well in straightforward scenarios, they are not suitable for the complex and dynamic nature of IoMT. For example, RBAC assigns permissions based on predefined roles, such as ‘doctor’ or ‘patient’, but it cannot adjust access dynamically based on real-time factors. If a user’s situation changes, e.g., a doctor being in a specific location within a hospital or handling an emergency, the system cannot automatically grant additional permissions to accommodate these new contextual demands. Additionally, DAC, while empowering users to control access to their data, lacks the *granularity* or fine level of control required for environments with a large number of interconnected devices and users, like in IoMT.

Moreover, centralized systems for managing access control in IoMT rely on a single point of control, such as a hospital’s internal server or cloud infrastructure, to store and manage patient data. This centralization creates a potential single point of failure—if the central authority is compromised, it can expose a vast trove of sensitive data to attackers, as exemplified by the 2017 NHS cyberattack. Additionally, centralized systems struggle with transparency and accountability. In many cases, it is difficult to trace who accessed or modified specific data. Patients often have limited insight into the handling of their data and may question whether their data is being accessed appropriately, raising concerns about data privacy and security within the IoMT system.

**Our approach:** Our research addresses these critical issues by proposing a blockchain-based medical data exchange system, MedAccessX, tailored for IoMT networks. This system leverages the decentralized and immutable features of blockchain to ensure secure, transparent, and auditable access to sensitive patient data across various healthcare providers. The core innovation of our approach is the seamless integration of *attribute-based access control* and *role-based access control* within smart contracts, enabling dynamic and fine-grained control over data access.

In our system, attribute-based access control (ABAC) allows access permissions to be determined by real-time attributes, such as a user’s location, role, or current medical emergency status. This dynamic adaptability is important in healthcare settings, where access needs can shift rapidly. For example, during an emergency, a doctor’s location or a patient’s immediate medical needs might require temporary access to critical data, which our system can facilitate efficiently. On the other hand, RBAC provides role-specific permissions, ensuring that healthcare professionals, such as doctors, nurses, or patients, access only the data pertinent to their roles. Combining these two methods in a hybrid model allows for a more flexible framework that adjusts access based on both static roles and real-time contextual factors, making it significantly more suitable for the dynamic and distributed nature of IoMT environments.

Another key advantage of our system is that every data access request is immutably logged on the blockchain. This feature provides an unparalleled level of transparency and accountability, ensuring that every interaction with patient data can be traced and verified. The innovation here is that, unlike traditional systems where audit logs can be stored in centralized, potentially vulnerable databases, our approach guarantees that all access logs are distributed and tamper-proof. This significantly enhances trust among healthcare providers, patients, and regulators.

**Contributions.** Our contributions are summarized as follows:We introduce a new blockchain-based medical data exchange system, MedAccessX, tailored for IoMT networks. Our system innovatively integrates ABAC and RBAC within smart contracts, enabling dynamic access control while achieving privacy-preserving, security, and auditable data sharing across healthcare providers.We conduct experiments to evaluate system performance. Our experiments measure key metrics such as smart contract execution cost, system throughput, average response time, and the efficiency of access control mechanisms. These solid results demonstrate the system’s scalability and reliability under various conditions.We compare MedAccessX with the state-of-the-art access control mechanisms and demonstrate that our approach is more cost-effective.We have open-sourced our code and results on GitHub (Codes of implementations are released at https://github.com/Minfeng-Qi/medicalDataDapp, accessed on 13 March 2025), providing transparency and enabling re-productivity.

In the following sections, Section 2 covers the foundational concepts that support our approach. In Section 3, we introduce our blockchain-based dynamic access control framework. Section 4 evaluates the system’s protection against threats, while Section 5 presents experimental results, comparing our method with existing solutions. Section 6 discusses related work, reviewing existing access control methods in different scenarios and their limitations. We conclude this paper and highlight potential avenues for future research in Section 7.

## 2. Preliminaries and Definitions

### 2.1. Attribute-Based Access Control

The ABAC model is an attribute-based access control method [7]. It implements access control by evaluating the attributes of the subject and object, as well as predefined policy rules. ABAC’s formal definition is as follows.

**Definition 1** (Policy Set)**.** 
*A policy set P is*

(1)
$$P=\{p_{i}|p_{i}:2^{SA}\times 2^{OA}\to \left\{\mathrm{ture},\mathrm{false}\right\}\},$$

*where S and O denote the Subject Set and Object Set, respectively, while SA and OA represent the attributes of subjects and objects. The subject attribute function SA is defined as $S\to 2^{SA}$, mapping the subject set S to subsets of their attribute set $2^{SA}$. Similarly, the object attribute function OA is defined as $O\to 2^{OA}$, mapping objects O to subsets of their attribute set $2^{OA}$. Here, $2^{SA}$ and $2^{OA}$ are the power sets of subject and object attributes, respectively. $2^{SA}\times 2^{OA}$ denotes the Cartesian product of these two power sets.*

*Each policy $p_{i}$ is a mapping that maps a combination of sets $2^{SA}\times 2^{OA}$ to either true or false.*

*An access request (AR) is represented as a tuple AR=(s,o), where s∈S and o∈O.*


**Definition 2** (Access Decision Function)**.** 
*The access decision function is the core component of ABAC, which determines whether to grant access rights based on the attributes of the subject, object, and environment, as well as a predefined set of policies. The function is as follows:*

(2)
$$\mathrm{Access}(a,o,p)=\left\{\begin{array}{cc} “allow”, & \mathrm{if}\;p_{i}=\mathrm{true};\\ “deny”, & \mathrm{otherwise}. \end{array}\right.$$



ABAC achieves flexible, dynamic, and fine-grained access control through a comprehensive evaluation of subject and object attributes [8]. It is suitable for systems that require complex permission management and real-time policy adjustment and can effectively improve the security, management efficiency, and compliance of the system. At the same time, the high flexibility and scalability of the ABAC model make it an ideal solution for applications in the blockchain field.

Although the ABAC model provides a high degree of flexibility and fine-grained control, it also has some shortcomings. First, as the number of attributes and policies increases, the complexity of the system escalates, making management and maintenance difficult, which may lead to configuration errors and increase security risks. In addition, real-time evaluation of multiple attributes and policies can impose a significant computational burden on the system, affecting performance, especially in high-concurrency environments. Thus, we shift our focus to the RBAC model, which offers an alternative approach to address these concerns.

### 2.2. Role-Based Access Control

The RBAC model is an access control model that associates permissions with specific roles and assigns users to these roles [9]. The RBAC formula is the following.

**Definition 3** (Core Relationships)**.** 
*In the RBAC model, the core relationships are the User-Role Assignment (UA) and the Role-Permission Assignment (PA). These two relationships define the mappings between users, roles, and permissions and are crucial components of the RBAC model. Below is a detailed explanation of these two relationships.*

**User-Role Assignment (UA)**

(3)
UA⊆U→R,


**Role-Permission Assignment (PA)**

(4)
PA⊆R→P,


*where U is the users set, R represents the set of roles, and P means the permissions set, and → is mapping relationship.*


Through UA and PA, the system can associate users with roles, determine each user’s roles, and assign corresponding permissions to each role.

**Definition 4** (Access Decision Function)**.** 
*When user u requests to execute a permission p, the system checks whether there exists a role r that has been assigned to the user u and has been granted the permission p. The function is*

(5)
$$\mathrm{Access}(u,r,p)=\left\{\begin{array}{cc} “allow”, & \mathrm{if}\;(u,r)\in UA,(r,p)\in PA;\\ “deny”, & \mathrm{otherwise}, \end{array}\right.$$

*where u∈U,p∈P,r∈R.*


The RBAC model streamlines system access control by assigning permissions to roles and then allocating these roles to users, which greatly simplifies the complexity of the system permissions allocation.

While RBAC offers a structured access control approach, it does have its drawbacks. First, RBAC is based on static roles and cannot dynamically adjust permissions based on the environment or user attributes. Thus, it lacks support for fine-grained and contextualized access control. In addition, RBAC makes it difficult to handle temporary permissions. For temporary tasks or special circumstances, it is not convenient to grant or revoke permissions, which lacks flexibility. Finally, in large organizations or cross-organizational environments, unified role definition and management may be challenging, affecting the interoperability of the system.

By combining the RBAC model with the ABAC model, we have built a dynamic access control system that harnesses the advantages of simplicity and manageability of the RBAC model, complemented by the flexibility and fine-grained control capabilities offered by the ABAC model. This combination can reduce the cost of the access control solution on the basis of fully defining access rights.

### 2.3. Blockchain

Blockchain technology is a decentralized distributed ledger system with characteristics such as transparency, immutability, and high security [10]. In the blockchain network, a chain structure consisting of blocks is generated, and each block contains many transactions [11]. The following is the formal definition of a blockchain.

**Definition 5** (Blockchain)**.** 
*Blockchain (BC) is a collection of blocks linked in sequence.*

(6)
$$BC=\{B_{0},B_{1},B_{2},\ldots ,B_{n}\},$$

*where $B_{0}$ is the genesis block.*

*Each block $B_{i}$ is a tuple made by*

(7)
$$B_{i}=\left\{H_{i-1},D_{i},T_{i},N_{i}\right\},$$

*where $H_{i-1}$ denotes the hash value of the previous block. For the genesis block $B_{0}$, we define $H_{-1}=0$ (or a default value). The variable $D_{i}$ represents the data or transaction set of the current block, $T_{i}$ symbolizes the timestamp, and $N_{i}$ denotes the random number or nonce value.*


**Definition 6** (Transaction)**.** 
*The basic unit of operation in blockchain and smart contracts is the transaction. A transaction $t_{x}$ is a tuple*

(8)
$$t_{x}=\left\{A_{from},A_{to},V,D,T_{i},Sig\right\},$$

*where $A_{from}$ and $A_{to}$ denote the sender’s and receiver’s address, respectively; V represents the value or amount of the transfer (e.g., number of tokens); D is the additional data, such as the function name and parameters to be evoked by the contract; Sig is the digital signature, which validates the authenticity and integrity of the transaction, ensuring its legitimacy.*


**Definition 7** (Smart Contract)**.** 
*A smart contract (SC) is a piece of code or program deployed on the blockchain, which can be regarded as a state transition function:*

(9)
$$SC:\left(S,I\right)\to \left(S^{\prime },O\right),$$

*where S is the current state of the contract, I denotes the set of inputs (e.g., transactions $t_{x}$ or external data), $S^{\prime }$ represents the next state set of the contract, and O refers to the output set (such as event logs, transfers, etc.).*


Smart contracts can autonomously enforce predefined access control policies without human intervention [12], thereby minimizing human errors and enhancing system efficiency and reliability. Additionally, the decentralized and immutable nature of smart contracts facilitates auditing and tracking of permission changes, which strengthens the security and trustworthiness of the system.

To this end, we leverage smart contracts in conjunction with ABAC and RBAC to develop an efficient, secure, and flexible access control framework named MedAccessX.

## 3. Our Scheme: MedAccessX

This section details our fine-grained access control solution for medical data sharing in IoMT environments.

**Hybrid design:** MedAccessX leverages a hybrid approach that combines attribute-based access control and role-based access control. ABAC allows permissions to adjust dynamically based on real-time attributes, such as location, device status, or patient context, ensuring flexibility in rapidly changing healthcare situations. RBAC establishes baseline permissions according to predefined roles like doctor, nurse, or patient, ensuring foundational access control based on professional responsibilities. By embedding these mechanisms into smart contracts, MedAccessX automates dynamic access management, adapting permissions in real-time based on current conditions and user attributes while maintaining the scalability essential for large-scale IoMT networks.

**Workflow:** Our framework consists of two main processes: a secure access control mechanism driven by four types of smart contracts deployed on the blockchain and decentralized IoMT data storage via the blockchain-based InterPlanetary File System (IPFS) (see Figure 1).

First, IoMT users register through a client application to obtain a unique *userID*, which is saved in the User Management Contract (UMC) along with user attributes such as *userName*, *role*, and *department*.

Next, data generated by IoMT devices is stored on IPFS, which then uploads metadata (including *dataOwner*, *ipfsCID* (the identifier used to uniquely identify data content in IPFS), and *dataType*) to the medical data management contract (MDMC) on the blockchain.

To access data, users log in via the client using their *userID* and specify the data owner. The access control contract (ACC) verifies their permissions through predefined policies in the policy contract (PC). If access is granted, MDMC retrieves the relevant *ipfsCID* and sends it to IPFS, which returns the data to the client for download.

We present details for each contract below.

**Threat model:** In our system, we consider that adversaries can be either external attackers or malicious insiders with varying levels of access to the network and data.

External attackers are considered completely untrustworthy. They may use denial of service attacks (DoS) and distributed denial of service attacks (DDoS) to send a large number of requests to the proposed MedAccessX, causing server overload and network paralysis, making the system unable to provide services to legitimate users. Additionally, these attackers might conduct man-in-the-middle attacks (MITM), intercepting and potentially tampering with the communication between two entities to extract sensitive information. Phishing attacks are also a concern, where attackers, disguised as trusted entities, trick users into providing sensitive information, such as their login credentials *userID*, etc., to commit identity theft.

Internal attackers are semi-honest. Attackers legally have the authority to access part of the data but may be tempted to illegally upgrade their permissions to access additional data or to illegally sell user data to gain benefits.

We consider four types of threats as follows:*Unauthorized data access:* It is a critical threat where adversaries aim to access sensitive medical information without authorization. Attackers may exploit vulnerabilities within the access control mechanisms to bypass restrictions. For instance, an attacker could attempt to manipulate the *userID*, *role*, or other attributes managed by the UMC to impersonate legitimate users with higher access privileges, such as admins.*Data integrity attacks:* These attacks involve attempts by attackers to alter or delete stored medical data or to manipulate access control logs, thereby undermining the integrity of the access control system.*Denial of Service (DoS) attack:* This threat targets the availability of the MedAccessX system by flooding it with excessive access requests. An adversary may flood the ACC with multiple requests, consuming excessive computational resources and bandwidth, which could slow down response times or prevent legitimate users from accessing the MedAccessX system.*Sybil attack:* A Sybil attack is a threat to the blockchain system. An adversary creates multiple fake identities or nodes to gain disproportionate control over the network. In the context of MedAccessX, such an attack could allow the adversary to manipulate access control decisions, tamper with data policies, or disrupt the consensus process.

**Assumptions used:** Our framework operates under several assumptions to ensure secure access control in IoMT networks. First, we assume that the blockchain network is permissioned, implying that only trusted nodes, such as authorized healthcare institutions, can join the network and validate transactions. Second, in MedAccessX, hash encryption is performed when the information is transmitted. We also assume that while network communications are susceptible to interception, adversaries do not have the computational power to break standard cryptographic primitives.

### 3.1. User Management Contract (UMC)

UMC manages user identities within the IoMT network. Leveraging blockchain’s immutability, the UMC establishes a decentralized registry that securely records each user’s identity, ensuring uniqueness and integrity. Each user in the system is assigned a unique identifier (*userID*) generated through the keccak256 hash function. This function combines the user’s core attributes—username, address, role, and department—into a cryptographic hash, producing a secure and unique *userID* that prevents conflicts and ensures consistent user identification across the network.

UMC supports a suite of essential functions for handling user data dynamically. For instance, the addUser function generates a new user entry, assigning a unique identifier, *userID*, to a user structure containing fields for *userName*, *userAddress*, *role*, and *department* (the logic behind this function is detailed in Algorithm 1). The new entry is then stored within the UserList (Figure 2), which maintains an immutable record of users added to the system. Additionally, to confirm that the information has been correctly recorded, our system incorporates a verification mechanism within the function. This step checks that the user’s address matches the input parameter, ensuring data consistency. Upon successful addition, an event, *UserAdded*, is triggered, logging details such as *userID*, *userName*, *userAddress*, *role*, and *department*, which enables real-time monitoring of user changes. Users and administrators call the adduser function in UMC deployed on the blockchain on the client to register users and obtain a unique *userID*.

To accommodate changes in user roles or departments, the updateUser function provides a streamlined method for modifying existing user data. Upon invocation, it retrieves the user entry from UserList and confirms the user’s existence. Updated role or department fields are hashed to produce a new *userID*, which is stored within the UserList and replaces the previous entry. In addition, by taking the *userID* as input, the deleteUser function erases the specific entry from the UserList, resetting the user’s data to its default values.

The getUser function allows secure and efficient retrieval of user information based on the *userID*. This view-only function verifies the presence of the user by checking the length of the *userName* field. If the user is registered, the contract returns the corresponding User struct, enabling transparent access to user details without altering the blockchain state.

The getUserHash function enhances verification by generating a cryptographic hash of the user’s complete information. Upon receiving a *userID*, the function retrieves the user’s data fields—*userName*, *userAddress*, *role*, and *department*—and compiles them into a byte array. It then applies the keccak256 hash function to this array, producing a bytes32 hash value that uniquely represents the user’s information. This hash serves as a tamper-proof and verifiable reference to the user’s data, which can be used in auditing user records across the system.
**Algorithm 1** addUser Function**Input:** 
*userName*, *userAddress*, *role*, *department***Output:** 
*userID*1:Generate *userID* by hashing *userName*, *userAddress*, *role*, and *department* using keccak2562:Create a new user struct *newUser* with fields:3:   *userName*: userName4:   *userAddress*: userAddress5:   *role*: role6:   *department*: department7:Add *newUser* to the *UserList* mapping with key *userID*8:Update *lastUserID* to *userID*9:Emit event UserAdded (*userID*, *userName*, *userAddress*, *role*, *department*)10:**if** *users*[*userID*].*userAddress* not equal to *userAddress* **then**11:   Throw error “User not added correctly.”12:**end if**13:**return** 
*userID*

### 3.2. Medical Data Management Contract (MDMC)

MDMC enables secure storage and traceable management of medical data across decentralized IoMT networks. This smart contract establishes a mechanism for logging, identifying, and retrieving medical data entries, leveraging the traceability of blockchain to support privacy-preserving access.

Each medical data entry is assigned a unique identifier (*dataID*), generated through the keccak256 hash function, which combines the *dataType*, *dataOwner*, and *ipfsCID* (the identifier for data stored on IPFS). This unique identifier allows each data entry to be securely referenced without exposing sensitive information on the blockchain.

The primary data entry process is handled by the addData function (see Algorithm 2). Upon receiving the *dataType*, *dataOwner*, and *ipfsCID*, the function first generates a unique *dataID* via keccak256 hashing. To protect data confidentiality, the function uses an internal encryptCID function, which encrypts the *ipfsCID*, safeguarding the link to the data stored on IPFS. This encrypted identifier is then stored within the contract’s registry, allowing verifiable access while maintaining the privacy of sensitive data storage locations. The addData function will store the information of each data on the blockchain to facilitate data management by users.

The contract’s structure is designed to facilitate tracking and retrieval. Each data entry, encapsulated in a *MedicalData* instance, is mapped to its *dataID* and appended to a DataList array, ensuring organized access to all entries within the contract. Additionally, to promote transparency, the contract triggers a *LogMedicalData* event upon adding new data, which broadcasts key information (e.g., *dataType*, *dataID*, and *dataOwner*) to external systems. This event-driven design supports real-time data monitoring, an essential feature in compliance-focused IoMT applications.

For secure data retrieval, the getMedicalID function provides access to stored entries based on the unique *dataID*. This function first validates the existence of the requested record by checking the *dataType* field to confirm the entry’s presence. If valid, it retrieves and decrypts the *encryptedCID*, which is the encrypted identifier for the data stored on the IPFS, providing authorized access to the stored data without compromising data integrity.
**Algorithm 2** addData Function.**Input:** 
*dataOwner*, *dataType*, *ipfsCID***Output:** 
*dataID*1:Generate *dataID* by hashing *dataType*, *dataOwner*, and *ipfsCID* using keccak2562:Call internal function encryptCID (*ipfsCID*) to get *encryptedCID*3:Create a new *MedicalData* instance *newMedicalData* with fields:4:   *dataType*: dataType5:   *dataOwner*: dataOwner6:   *ipfsCID*: encryptedCID7:   *fileContent*: fileContent8:Store *newMedicalData* in the *medicalData* mapping with key *dataID*9:Create a new *DataRecord* instance *newDataRecord* with fields:10:   *dataType*: dataType11:   *dataOwner*: dataOwner12:   *ipfsCID*: encryptedCID13:   *dataID*: dataID14:Add *newDataRecord* to the *dataList* array15:Append *dataID* to *ownerToDataIDs*[*dataOwner*] in the mapping16:Emit event LogMedicalData(*dataID*, *dataType*, *dataOwner*, *encryptedCID*, *fileContent*)17:**return** 
*dataID*

### 3.3. Policy Contract (PC)

The policy contract (Figure 3) is a pivotal component, used to determine access permissions within the IoMT environment. This contract integrates both RBAC and ABAC models to address diverse access requirements. The selection of the access model is dynamically managed by a comprehensive decision function, hasAccess, which evaluates user attributes and roles to determine the appropriate access policy.

The hasAccess function is defined as follows:(10)$$\mathrm{HasAccess}(u,o)=\left\{\begin{array}{cc} \mathrm{Access}(u,r,p), & \mathrm{if}\;s=\mathrm{false};\\ \mathrm{Access}(a,o,p), & \mathrm{if}\;s=\mathrm{true}. \end{array}\right.$$

In this function, s is a boolean variable indicating whether the ABAC model should apply. When s=true, the function uses the ABAC model to evaluate access permissions with Access(a,o,p), as defined in Formula (2). When s is set to false, the function defaults to the RBAC model with Access(u,r,p), as described in Formula (5).

**RBAC in policy contract:** In the policy contract, RBAC serves as the foundational model for granting permissions based on predefined roles associated with each user. This approach simplifies access control by assigning permissions based on users’ roles. RBAC is the primary model for the majority of users whose access requirements are role-dependent.

The RBAC model operates by linking each role (e.g., doctor, nurse, technician, or patient) to a specific set of permissions for accessing different resources within the IoMT network. Permissions are defined according to the tasks or responsibilities associated with each role, ensuring that users can access the necessary resources to perform their duties while preventing unauthorized access to unrelated data.

Formally, access decisions in RBAC are evaluated using the function Access(u,r,p), where u represents the user, r represents the user’s role, and p is the requested resource. For instance, the function Access(u,Patient,PatientRecord) grants a patient access to their own medical records, while Access(u,Nurse,NursingRecord) enables a nurse to access nursing records within their department.

The simplicity of the RBAC model facilitates efficient authorization in scenarios where users’ access requirements are static and role-based. By associating permissions directly with roles, the policy contract can quickly authorize or deny requests based on role-based mappings without requiring complex attribute evaluations. However, for cases where more nuanced control is necessary, such as for administrative or specialist roles, the policy contract seamlessly integrates ABAC to support attribute-based access decisions, creating a hybrid model that balances simplicity with flexibility.

**Policy definition in ABAC:** Policies are defined as rule sets that evaluate access rights based on subject and object attributes (Table 1). Each policy rule p can be represented as:(11)p=(SA,OP,SV,OA,OV,Act).

In this formula, SA (Subject Attribute) denotes the attribute of the user, such as their role or department; OP (Operator) specifies the comparison operation (e.g., equality, inequality); SV (Subject Value) is the expected value of the subject attribute; OA (Object Attribute) is the attribute of the resource, such as data type or ownership; OV (Object Value) is the expected value of the object attribute; and Act is the action that determines whether the policy grants or denies access, set to either “allow” or “deny”.

**Policy condition evaluation:** For each policy p, we define a Condition Function Con(a,o,p) that determines whether an access request from user u for resource o satisfies the requirements of policy p. This function evaluates two main conditions: (i) The user’s attributes a match the specified conditions for the subject. (ii) The resource’s attributes o match the specified conditions for the object.(12)$$\begin{array}{cc} \mathrm{Con}(a,o,p)= & \;\mathrm{Comp}({\mathrm{Attr}}_{u}\left(u,SA\right),SV,OP)\\ \; & \wedge \mathrm{Comp}({\mathrm{Attr}}_{o}\left(o,OA\right),OV,“=”), \end{array}$$
where ${\mathrm{Attr}}_{u}\left(u,SA\right)$ retrieves the user attribute specified by SA from user u; ${\mathrm{Attr}}_{o}\left(o,OA\right)$ retrieves the resource attribute specified by OA from resource o; and Comp(a,b,op) is the Comparison Function that evaluates attribute values a and b according to the operator op. This function enables flexible attribute verification, supporting various conditions defined in the policy.(13)$$\left\{\begin{array}{cc} \mathrm{Comp}(a,b,“==”) & \to (a=b);\\ \mathrm{Comp}(a,b,“!=”) & \to (a\neq b);\\ \mathrm{Comp}(a,b,“\mathrm{contains}”) & \to (b\;\mathrm{is}\;a\;\mathrm{substring}\;\mathrm{of}\;a). \end{array}\right.$$**Access decision function:** The Access Decision Function Access(a,o,P) combines the results of individual ABAC policy evaluations to make a final access decision. Given a set of policies P, this function iterates through each policy p∈P and checks if any policy grants or denies access based on the conditions evaluated by Con(a,o,p).(14)$$\begin{array}{c} \mathrm{Access}\left(a,o,p\right)=\left\{\begin{array}{cc} \left(i\right)\;\mathrm{true}, & \mathrm{if}\;p\in P,\mathrm{Con}(a,o,p)\\ \; & \;\wedge p.\mathrm{Act}=“allow”;\\ \left(\mathrm{ii}\right)\;\mathrm{false}, & \mathrm{if}\;p\in P,\mathrm{Con}(a,o,p)\\ \; & \;\wedge p.\mathrm{Act}=“deny”;\\ \left(\mathrm{iii}\right)\;\mathrm{false}, & \mathrm{if}\;\mathrm{no}\;\mathrm{matching}\;\mathrm{policy}, \end{array}\right. \end{array}$$
where (i) If there exists a policy p in P such that Con(a,o,p) returns true and the policy action p.Act is set to “allow”, then access is granted; (ii) If there exists a policy p in P such that Con(a,o,p) returns true and the policy action p.Act is set to “deny”, then access is denied; (iii) If there is no policy in P matched, access is denied by default.

**Policy evaluation process:** The policy evaluation process involves checking all defined policies to determine if any of them grant access to the user for the requested resource. Access is granted if at least one policy condition is met.

The process iterates over the set of policies *P*, evaluating each policy condition Con(a,o,p) for the user and resource. This is represented as follows:(15)$$\mathrm{Access}\left(a,o,p\right)=R_{1}\vee R_{2}\vee R_{3}\vee \cdots \vee R_{n}.$$

In this equation, $R_{1},R_{2},R_{3},\cdots ,R_{n}$ represent individual policies that satisfy the condition Con(a,o,p)=true. The symbol ∨ denotes the logical OR operator, indicating that if any policies in the set *P* meet the access condition, access will be granted.

The evaluation proceeds as follows:*Condition check*: For each policy p∈P, the system evaluates the condition Con(a,o,p). If the condition is true and the policy action p.Act is “allow,” that policy grants access, and the evaluation process can terminate early with a positive result.*Logical aggregation*: The logical OR (∨) operation across policies ensures that access is granted if at least one policy’s condition is met with an “allow” action.*Default deny*: If no policies satisfy Con(a,o,p)=true with an “allow” action, access is denied by default. This ensures a secure stance, as access is only granted if explicitly allowed by at least one policy.

### 3.4. Access Control Contract (ACC)

ACC, as the core component, coordinates with the User Management Contract (UMC), medical data management contract (MDMC), and policy contract (PC) to execute the access control process seamlessly.

**Access control workflow:** The ACC’s access control logic (Algorithm 3) follows a sequence of steps to verify user credentials, evaluate permissions, and retrieve authorized data. This process involves a combination of user authentication, permission checking, and data retrieval, leveraging the other contracts. Here’s a breakdown of the key steps:
**Algorithm 3** loginUser and AccessData Function**Input:** 
*userID*, *dataOwner*, *dataType***Output:** 
*false*, *Data*1:Retrieves user information from *UserManagement* using *userID*2:**if** user exists **then**3:   Update *userSessions* with *isLoggedIn*, *role*, and *department*4:   Check access permissions from PC5:   **if** access is granted **then**6:     **return** result from accessDataByOwner(*dataOwner*, *dataType*)7:   **else**8:     emit LogAccessDenied(*msg.sender*, “Access denied for the requested data type.”)9:     **return** *false*10:   **end if**11:**else**12:   Emit LogAccessDenied event with the reason of “User does not exist.”13:   **return** *false*14:**end if**

*User authentication*: The process begins with the loginUserAndAccessData function, which verifies a user’s identity based on their unique *userID*. By invoking the getUser function from the UMC, ACC retrieves the user’s role and department information, using it to update the user’s session data. The result of this session update is stored in userSessions[msg.sender], indicating the user’s current login status along with their role and department:(16)U=UMC.getUser(userID).If the user exists, their session is updated:(17)$$U_{s}\left[msg.sender\right]=\left(\mathrm{true},U.role,U.department\right).$$*Permission verification*: With the session information in place, ACC proceeds to verify the user’s access permissions by invoking the hasAccess function within the PC. This function checks if the user’s attributes (role, department) satisfy the required access conditions for the requested data. The verification step is(18)A(r,d,o,t)←hasAccess(u,o),
where r is the user’s role, d represents the user’s department, o is the data owner, and t is the data type. If permission is granted, ACC proceeds to retrieve the data. Otherwise, an access-denied response is triggered, logging the denial event with a message such as “Access denied for the requested data type.”*Data retrieval*: Upon authorization, data retrieval from MDMC by the accessDataByOwner function focuses on data owned by the specified data owner and filtered by data type. The ACC first retrieves all data identifiers associated with the data owner:(19)D=MDMC.getUserDataIDs(dataOwner).ACC then traverses this set D, selecting only those data entries that match the requested data type. For each matching identifier $d_{i}$, the corresponding medical data $M_{D}\left(d_{i}\right)$ is retrieved:(20)$$M_{D}\left(d_{i}\right)=\mathrm{MDMC}.getMedicalData\left(d_{i}\right).$$If a matching data type is found, ACC converts the *ipfsCID* to a string for output and triggers the *LogDataAccessed* event to record the access operation. If no matching records are found, a “false” response is returned, indicating that no data met the requested type.

**Practical access control process:** Figure 4 outlines the complete flow of the access control process:

*Step 1: Registration and storage:* Users register through a web-based interface, where their encrypted registration information generates a unique *userID*. This identifier is recorded in the UMC and stored within the *UserList*. Separately, when users upload data to IPFS, the CID generated by IPFS is sent to MDMC, where it is added to the *DataList*.

*Step 2: User login and authorization:* During login, the user enters their *userID*, which ACC verifies against the *UserList*. Based on the user’s role and department, ACC consults the PC to confirm if the user has the appropriate permissions to view the requested data type. If unauthorized, ACC returns a response, indicating “Access denied, authorization required”.

*Step 3: Data access and retrieval:* If authorized, ACC queries the *DataList* in MDMC, retrieving only those records that match the user’s access rights. The data is then made available for download via the web interface.

## 4. Security Analysis

In this section, we evaluate the security aspects of MedAccessX, mainly on how MedAccessX avoids these attacks.

**DoS attack:** The attacker consumes system resources through a large number of invalid requests [13], making the system unable to provide services normally. MedAccessX mitigates this threat by leveraging a combination of on-chain and off-chain measures. For example, the rate-limiting mechanism is implemented in the ACC to limit the number of requests from each user within a specified timeframe, reducing the impact of flooding attacks. Additionally, MedAccessX’s decentralized structure, with data stored on IPFS and access control managed through smart contracts on the blockchain, enhances resilience against DoS attacks.

**Sybil attack:** An attacker creates a large number of fraudulent identities to increase his influence in the network, thereby manipulating the network or disrupting its normal operation [14,15]. MedAccessX mitigates Sybil’s attacks through multiple layers of defense: it operates on a permissioned blockchain with strict identity verification, ensuring only authorized entities can participate. RBAC and ABAC enforce policies that limit unauthorized access, while all actions are logged immutably on the blockchain for traceability.

**Transaction order dependency attack:** The attacker takes advantage of the control of transaction ordering by blockchain miners when producing blocks and seeks economic benefits by adjusting the execution order of transactions [16]. In our solution, the operations in the four contracts are independent of the transaction order, and there is no race condition, thus preventing any transaction order dependency attack.

**Inference attack:** Attackers infer sensitive information by analyzing existing public or non-sensitive data, threatening the user’s data security [17]. In MedAccessX, the contract only discloses necessary metadata to avoid exposing too much information on the chain that may be used for inference. Due to the openness and transparency of the blockchain, it is impossible to completely hide the access mode. However, by refraining from storing sensitive data on the chain, the possibility of inferring sensitive information is reduced.

## 5. Performance Analysis

In this section, we conduct a comprehensive evaluation of MedAccessX. We analyze the deployment costs, average gas consumption, throughput, and average response time. We also conduct comparisons.

### 5.1. Experimental Settings

To simulate the blockchain environment, we utilize Ganache [18] for local blockchain simulation and deploy our framework on a local server equipped with an AMD Ryzen 5 5600X 6-core processor produced by Advanced Micro Devices, Inc. (AMD) headquartered in Santa Clara, CA, USA. at 3.70 GHz and 16 GB of memory. For performance testing, we set a maximum of 2000 transactions, beginning with the first transaction (skipping the 0th) and measuring throughput every 20 transactions until the final count of 2000. Given that real IoMT scenarios often involve concurrent access by multiple users, we simulate varying levels of concurrency by configuring one, five, ten, and twenty nodes, enabling us to assess the framework’s performance in multi-user settings.

**Key functions settings:** Our performance analysis focuses on three primary functions: addUser, addData, and accessControl. We describe detailed function configurations.

-For the addUser function, we randomly generate user attributes such as Name, hashAddress, Role, and Department. Each user registration is recorded, capturing metrics including average response time, throughput, and gas consumption. This data enables us to compare resource usage across different access control models.-The addData function test follows a similar setup in the addUser function test, where attributes like dataOwner, ipfsCID, and dataType are randomly generated. We measure performance metrics across multiple iterations to ensure accuracy, following similar steps as in the addUser function test.-Testing the accessControl function is more complex due to the need to handle multiple user roles and varying permission levels. For this test, we first register 30 users for each role, i.e., doctors, nurses, and patients, and store their information within the contract. We then generate a pool of registered and unregistered userIDs, shuffle them, and store them in a user information file. This file is used to simulate access attempts, yielding three possible outcomes: failed login, successful login with insufficient permissions, and successful login with data access. Each result is logged in a CSV file, providing a comprehensive dataset for analyzing access control effectiveness.

### 5.2. Deployment Cost Discussion

**Transaction fee:** Deploying and executing smart contracts incur transaction fees, which are determined by gas consumption and the gas price [19]. In Ethereum, gas measures the computational complexity of a transaction, with more complex operations requiring higher gas amounts. Like gasoline at a gas station, gas in Ethereum has a variable price that can fluctuate based on network demand. The transaction fee (*TXFee*) for executing a smart contract can be calculated as follows:(21)$$\mathrm{TXFee}=gasUsed\times gasPrice\times 10^{-9},$$
where the unit of *TXFee* is *Ether*, and the *GasPrice* is assumed to be 20 Gwei.

**Deployment cost:** We measured and discussed the deployment costs of each contract.

-*User management contract:* The User Management Contract handles functions such as *addUser*, *deleteUser*, and *updateUser*. Due to its reliance on complex data structures (e.g., *structs*) and mappings and the need for multiple data operations, UMC has a relatively large bytecode size. This complexity results in a higher deployment cost, with a gas usage of 820,006 Gwei, corresponding to approximately $40.97. The user attributes like *userName*, *userAddress*, *role*, and *department* contribute to this higher cost, as each attribute increases the demands for storage and computation.-*Medical data management contract:* The medical data management contract primarily manages the storage and retrieval of medical data. Although MDMC shares similar data structures and mappings with UMC, its functionality is more streamlined, focusing on data storage and retrieval in DataList rather than complex user management. This results in a reduced bytecode size and a comparatively lower deployment cost of 775,320 Gwei, or approximately $38.74. However, the need to secure sensitive medical data requires additional security measures, which may contribute slightly to the cost but remain lower than UMC due to MDMC’s less complex operations.-*Policy contract:* The policy contract defines access rights for various user roles, mainly containing permission-checking functions without extensive data structures or complex operations. With simpler logic, PC has the smallest bytecode size among the four contracts, resulting in the lowest deployment cost at 588,100 Gwei (approximately $29.36). Although the PC has a straightforward structure, it plays a pivotal role in access control; any inaccuracies in permission checks could compromise data access management.-*Access Control Contract:* The access control contract integrates functionalities from UMC, MDMC, and PC to implement user login and comprehensive access control. Due to the need to interact with multiple contracts and handle complex permission and session management, ACC has the largest bytecode size among the four contracts, leading to a deployment cost of 913,539 Gwei, or roughly $45.61. This cost is significantly higher than the other contracts because ACC’s complexity involves calling external contracts and verifying contract addresses, adding layers of computation and storage requirements.

The total deployment cost for all four contracts amounts to 3,096,965 Gwei (approximately $154.65), as summarized in Table 2. This breakdown reflects the varying complexities of each contract, with UMC and ACC incurring the highest costs due to their intricate data management and multi-contract interactions. In contrast, the MDMC and PC remain more cost-effective due to their simpler logic and data structures.

### 5.3. Average Response Time

The average response time (ART) refers to the time required for the system to complete processing a request and return the result, which reflects the efficiency of the system in processing requests. A lower ART means that the system can complete the operation faster.

Figure 5a–d illustrate the ART of the three primary functions, i.e., *addUser*, *addData*, and *accessControl*, across various node configurations (one, five, ten, and twenty nodes). The yellow curve represents the *addUser* function, the green curve shows the *addData* function, and the purple curve displays the *accessControl* function.

Overall, the ART for *addUser* is the lowest, while *accessControl* has the highest ART. This result aligns with the complexity inherent in each function. The *addUser* function requires minimal computation and primarily involves basic data storage in the *UserList*, making it efficient. In contrast, *addData* requires additional interaction with IPFS, including receiving and processing the CID, which increases processing time. The *accessControl* function has the highest ART due to the multi-step process of verifying user identity with UMC, checking permissions with PC, and accessing data via MDMC, which involves CID retrieval and decompression.

**Analysis across different node configurations:** As node numbers increase, ART rises across all functions, indicating that more simultaneous access requests challenge system capacity. For *accessControl*, ART fluctuations are most pronounced. This is likely due to varying scenarios in access requests: invalid userID, valid userID without data access rights, and valid userID with data access rights. Each scenario imposes a different load on the access control function, with the last scenario requiring the most resources, thus leading to higher ART and variability. The *addData* function also experiences greater ART fluctuations than *addUser*, as its processing requires more complex operations.

In the *1-node* configuration (Figure 5a), ART remains relatively stable, with minor increases over time as the number of transactions grows. This configuration indicates that with limited nodes, the MedAccessX system can handle requests efficiently with minimal fluctuations.

In the *five-node* configuration (Figure 5b), ART begins to exhibit mild fluctuations for *addData* and *accessControl*, especially as the transaction count approaches 2000. This suggests that MedAccessX can maintain reasonable performance under moderate load, but additional nodes introduce variability.

In the *10-node* configuration (Figure 5c), ART increases more noticeably, particularly for *accessControl*, with more significant fluctuations. This configuration shows that while the system can handle an increased number of transactions, ART stability decreases, especially for more complex operations.

In the *20-node* configuration (Figure 5d), ART reaches its peak, with significant increases and fluctuations, especially for *accessControl*. This suggests that the system approaches its maximum load capacity under high concurrency, resulting in an ART increase and variability, which reflects the system’s stress under such conditions.

**Insights and key takeaways:** The ART differences among *addUser*, *addData*, and *accessControl* highlight the impact of function complexity on response time. The function *accessControl*, with its multi-layered verifications, has the highest ART, followed by *addData* and *addUser*. In addition, ART increases with more nodes due to the increased number of simultaneous transactions. The fluctuations in ART are also more pronounced with higher node counts, particularly for *accessControl*, indicating that the system’s concurrency capacity directly influences performance consistency. Moreover, ART fluctuations, especially in the 20-node scenario, suggest that MedAccessX operates efficiently under moderate concurrency but reaches a stress threshold with a high number of nodes. The stability of *addUser* across different configurations suggests that simple operations are less impacted by node count. Furthermore, among the tested configurations, the 10-node setup balances performance and stability effectively, as ART fluctuations stabilize beyond the 200-transaction mark for *addUser* and *addData*. However, for higher loads, ART variability becomes more prominent, especially for complex operations.

### 5.4. Throughput

Throughput is a critical metric for evaluating the performance and processing capacity of blockchain-based systems, especially under high-concurrency conditions. It represents the number of transactions a system can process per unit of time, typically measured in transactions per second (TPS) [20]. Throughput in blockchain systems is influenced by several factors, including block size and time, consensus mechanism, network latency, bandwidth, and node performance.

Figure 6a–d illustrate the throughput trends of the *addUser*, *addData*, and *accessControl* functions as the transaction volume increases across different node configurations (one, five, ten, and twenty nodes). Throughput and ART are interdependent metrics; generally, systems with lower ART can sustain higher throughput under optimal resource availability.

With an increasing transaction load, throughput decreases for all three functions—*addUser*, *addData*, and *accessControl*—regardless of node count. This is anticipated, as an increase in the transaction volumes places a greater demand on system resources, causing a reduction in processing efficiency over time. The decline rates in throughput vary among the three functions, aligning with the previously observed ART trends. Specifically, the throughput curves exhibit the steepest decline for *accessControl*, followed by *addData*, and then *addUser*, which is related to the function’s complexity.

**Function-specific observations:** we also investigate the performance of each function.

-*AddUser function:* The *addUser* function shows the highest throughput, as it involves simpler operations (basic data storage without complex permissions or data retrieval). Interestingly, the throughput of *addUser* decreases gradually as node count increases, indicating that this function is less sensitive to node concurrency than to overall system load. In the interval of [200, 2000] transactions, the throughput of *addUser* under the 1-node setup shows a steeper decline compared with configurations with more nodes, highlighting its sensitivity to system load rather than node count.-*addData function:* The throughput of *addData* varies more significantly with node count. Notably, the highest throughput is achieved with 10 nodes, indicating that this configuration optimally balances load distribution and transaction processing speed for data addition tasks. This result suggests that when 10 users (or accounts) concurrently initiate data transactions, the system achieves peak efficiency for *addData*, surpassing the performance of the one, five, and twenty-node configurations.-*accessControl function.* The *accessControl* function shows a downward trend in throughput with varying levels of fluctuation across different node configurations. For node counts of one, five, and ten, the throughput exhibits fluctuations within the initial transaction interval [0, 300], with the five-node configuration showing the least variability. However, when the node count is increased to 20, throughput for *accessControl* becomes highly erratic, indicating that the system is reaching its maximum load capacity. The increased fluctuations are primarily attributed to the complexity of access control operations, which involve multiple verifications (e.g., identity verification and permission checks), compounded by concurrent requests. The random selection of userIDs (valid, invalid, or with/without access permissions) also contributes to this variability, but the primary cause is the system strain under peak load conditions.

**Insights and key takeaways.** The decrease in throughput across functions with increasing transaction count highlights the impact of system load. As load increases, the system’s processing efficiency declines, with the steepest reductions observed in the most complex functions like *accessControl*. Additionally, the throughput trends for *addData* and *accessControl* reveal that moderate concurrency (e.g., 10 nodes) can improve system performance by striking a balance in transaction processing. This finding suggests that our framework benefits from moderate node configurations, which prevent both resource underutilization and overloading. Furthermore, among the configurations tested, the 10-node setup achieves a favorable balance for throughput, particularly for *addData*. This setup maximizes system efficiency for concurrent transaction processing, indicating that the framework operates optimally under moderate concurrency levels.

### 5.5. Average Gas Cost

Average gas consumption (AGC) provides insight into the resource requirements for each transaction executed on the blockchain. In blockchain networks, gas represents the computing resources consumed by smart contract functions [21,22]. Operations like computation, storage, and data transmission all incur gas costs. AGC reflects the average amount of gas used per transaction, offering a measure for the complexity and resource intensity of each function. Key factors influencing gas consumption include contract complexity, the need for data storage and retrieval, and the frequency of function calls.

Figure 7a–d illustrate the AGC trends for the *addUser*, *addData*, and *accessControl* functions across varying node configurations (one, five, ten, and twenty nodes). We first analyze the gas cost formulas for each function and then discuss AGC behavior as the total transaction count increases.

**Function-specific gas cost analysis:** we conduct a breakdown evaluation for each function.

-*AddUser function:* The *addUser* function hashes the user’s basic information using the *‘keccak256’* hash function to generate a unique *‘UserID’*. This *‘UserID’* serves as the key, with the user information stored as the corresponding value in the UserList mapping. The function ultimately returns the newly generated *‘UserID’*. The cost of addUser $G_{addUser}$ is(22)$$G_{addUser}=\left(G_{b}+G_{su}+G_{eu}+G_{h}+G_{misc}\right)\times gasPrice,$$
where $G_{b}$ is the basic transaction costs, valued at 2100 gas.$G_{su}$ denotes the storage operation costs, calculated as follows:(23)$$G_{su}=x\times 4000+y\times 5000+z\times 2000,$$
where x represents the cost of new storage slots, y denotes the storage modification cost, and z signifies the cost of adding elements to the UserList array.$G_{eu}$ delineates event log costs, which consist of the cost of index parameters ($C_{i}$) and the cost of non-index parameters ($C_{ni}$), detailed by:(24)$$G_{eu}=C_{i}+C_{ni}=a\times 375+\left(b+l\right)\times 8,$$
where variables a and b correspond to the index and non-index parameters and the number of bytes of index and non-index data, respectively. The length of each string is defined by l. $G_{h}$ is the hash calculation cost given by the following equation:(25)$$G_{h}=30+6\times l.$$$G_{misc}$ represents other miscellaneous operating costs.-*-addData function:* The *addData* function is used to store medical data within the contract and generate a unique *‘dataID’*. We can use an analysis method similar to the addUser function. The total gas cost of the addData function $G_{addData}$ can be expressed as:(26)$$\begin{array}{cc} G_{addData} & =(G_{b}+G_{h}+G_{l}+G_{sd}\\ \; & \;+G_{ed}+G_{misc})\times gasPrice, \end{array}$$
where $G_{b},G_{h}$, and $G_{misc}$ represents the basic transaction costs, hashing costs, and miscellaneous operating costs, respectively, consistent with those in the addUser function.Additionally, $G_{l}$ denotes the cost of encrypting ipfsCID:(27)$$G_{l}=G_{lpi}\times l_{CID},$$
where $G_{lpi}$ is the cycle operating costs, and $l_{CID}$ denotes the loop through the byte length of *ipfsCID*.$G_{sd}$ and $G_{ed}$ denote the storage operation costs and event log costs, respectively.-*accessControl function:* The AC function implements the *accessControl* function by managing user sessions and verifying permissions. When a user invokes the *‘loginUserAndAccessData’* function, the contract first retrieves user information from UMC’s UserList using the provided *‘userID’* to ensure that the user exists. It then stores the user’s session information in the *‘userSessions’* mapping. Based on the user information, AC calls the corresponding function in the PC to check whether the user has permission to access the requested data type. If the user has access rights, the contract calls the ‘accessDataByOwner’ function to obtain data related to *‘dataOwner’* from the MDMC, matches the data type, and returns the corresponding *ipfsCID*. At the same time, the contract triggers events to record the user’s login and data access operations. If the user is either non-existent or lacks sufficient permissions, the contract triggers the *‘LogAccessDenied’* event and returns “false”. The gas cost of the *accessControl* function $G_{ac}$ is(28)$$\begin{array}{cc} G_{ac} & =(G_{b}+G_{gU}+G_{suc}+G_{ach}\\ \; & \;+G_{ad}+G_{eac})\times gasPrice, \end{array}$$
where $G_{b}$ represents basic transaction costs.$G_{gU}$ represents the cost of calling the UMC to retrieve user information:(29)$$G_{gU}=G_{call\mathrm{UMC}}+rf\times 2100,$$
where $G_{call\mathrm{UMC}}$ is the gas cost of calling the UMC and rf represents the number of read fields.$G_{suc}$ denotes the cost of storing user session data:(30)$$G_{suc}=z\times 2000,$$
where z is the number of storage slots.$G_{ach}$ symbolizes the cost of permission verification (calling the PC):(31)$$G_{ach}=G_{call\mathrm{PC}}+G_{policycheck},$$
where $G_{call\mathrm{PC}}$ is the cost of calling the PC, and $G_{policycheck}$ is the cost of policy check.$G_{ad}$ represents the cost of accessing MDMC to obtain data:(32)$$G_{ad}=G_{call\mathrm{MDMC}}+G_{getData},$$
where $G_{call\mathrm{MDMC}}$ is the cost of calling the MDMC, and $G_{getlData}$ is the cost of retrieving the data.$G_{eac}$ denotes the event cost of the access function, similar to those in the addUser function.

**Insights and key takeaways:** As shown in Figure 7a–d, AGC for all functions decreases with an increasing number of transactions. The initial AGC is high due to the fixed costs associated with establishing the initial transactions. However, as transaction volume increases, the AGC stabilizes, exhibiting an exponential decay pattern.

In the initial transaction interval [0, 180], the AGC shows a sharp decline across all functions. Beyond this range, specifically between [180, 2000], AGC stabilizes, with minor fluctuations. Notably, the AGC of the *accessControl* function in the *one-node* configuration (Figure 7a) exhibits significant fluctuations in this stable interval. This variability arises from the randomized selection of userIDs (e.g., invalid userID, valid userID with or without access rights), which leads to differences in gas usage per access scenario.

Across all configurations, the AGC of the three functions follows a clear pattern, with *accessControl* consuming the most gas, followed by *addData* and *addUser*. This ordering reflects the operational complexity of each function, as *accessControl* requires multiple inter-contract calls and permission verifications, whereas *addUser* involves only basic storage operations. Additionally, under different node configurations (one, five, ten, and twenty nodes), the AGC pattern remains consistent, though higher node counts result in slight increases in AGC due to concurrent processing overhead.

### 5.6. Comparison of ABAC and RBAC

Figure 8a,b show the comparison of average gas consumption and average response time between the two access control policy models (i.e., ABAC and RBAC) under both single-node and multi-node scenarios. The purple curve represents ABAC, and the green curve represents RBAC.

**Analysis of average gas consumption:** As illustrated in Figure 8a,b, both ABAC and RBAC models exhibit a decrease in AGC as the total number of transactions increases, with AGC stabilizing around 250 transactions. However, the AGC for the ABAC model is substantially higher than that of the RBAC model across all scenarios. This discrepancy is primarily due to the structural differences between ABAC and RBAC. The ABAC model requires traversing a greater number of policies to evaluate permissions, as each access request may involve complex, attribute-based conditions. In contrast, RBAC relies on predefined roles, making it more straightforward and less computationally intensive.

Furthermore, the AGC of ABAC exhibits higher variability compared with RBAC. This fluctuation in ABAC can be attributed to the dynamic nature of attribute-based policies, where the number of policies traversed varies for each access request. This variability suggests that optimizing the ABAC policy model could help reduce both gas consumption and processing time by minimizing unnecessary policy checks.

**Analysis of average response time:** Figure 9a,b illustrate the ART for both ABAC and RBAC models. Consistently, ABAC incurs a significantly higher ART than RBAC. This is anticipated, as ABAC’s flexible policy framework requires more computational resources to evaluate complex attribute-based conditions. RBAC, on the other hand, leverages fixed roles that streamline access checks, thereby resulting in a lower and more stable ART. Notably, ART for both models tends to increase as the number of transactions grows, with ABAC exhibiting larger fluctuations.

### 5.7. Comparison with Other Methods

To highlight the advantages of our solution, we compare it with existing blockchain-based access control solutions in terms of deployment cost and computational cost.

**Deployment cost comparison:** As shown in Figure 10, our solution exhibits a significantly lower deployment cost compared with other blockchain-based access control methods, including those by Zhang et al. [23], Guo et al. [24], Sultana et al. [25], and Wang et al. [26]. Specifically, our deployment cost of 3,096,965 gas is substantially lower than Wang’s solution, which has the highest deployment cost at 5,535,960 gas. This reduction in deployment cost highlights the efficiency of our approach, which minimizes resource consumption and optimizes cost-effectiveness. Such a reduction is crucial for IoMT applications, where system scalability and economic viability are essential for widespread adoption.

**Computational cost comparison:** Our solution achieves the lowest computational cost (0.1 ms) among all the evaluated schemes in Figure 11. This efficiency surpasses even the solutions by Zhang et al. [23] and Luo et al. [27], which also implement the ABAC access decision model but incur higher computational costs (0.14 ms and 0.11 ms, respectively). Notably, our solution also outperforms the more complex schemes by Shi et al. [28] and Lyu et al. [29], which have higher computational costs (1.79 ms and 1.29 ms, respectively). This improvement indicates that our solution not only processes transactions more rapidly but also optimizes resource usage, making it highly suitable for IoMT environments where rapid data access and low latency are paramount.

**Figure 10 sensors-25-01857-f010:**
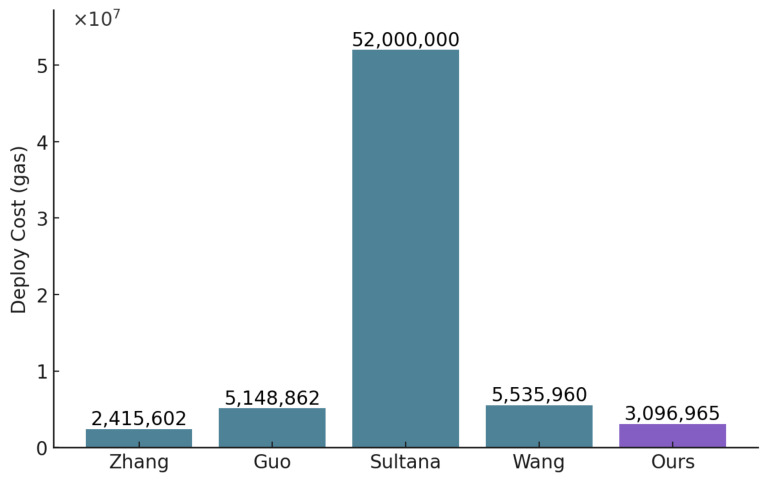
Comparison of the deployment cost with Zhang et al. [23], Guo et al. [24], Sultana et al. [25] and Wang et al. [26].

## 6. Related Work

This section presents related studies on access control models and provides a comparison (Table 3).

Zhang et al. [23] address the problems of single-point failure, low reliability, and poor scalability of centralized access control solutions in existing IoT and propose a smart city access control framework based on ABAC and SC. Based on this, our research adopts a combination of ABAC and RBAC when defining permission policies. By combining ABAC and RBAC, we reduce the number of permission settings, thereby reducing the gas consumption of the overall solution. Simulation results show that our solution yields lower computation and gas costs compared with existing solutions.

Aiming at the problems of data privacy protection, access control, and tracking and revocation of malicious users in IIoT, several researchers have put forward innovative solutions. Luo et al. [27] proposed a policy domain-based access control framework. Yu et al. [30] proposed a blockchain-enhanced secure access control scheme based on public–private key encryption that supports traceability and revocability. Wang et al. [26] proposed a dynamic IIoT access control management system based on smart contract tokens, innovatively converting traditional access control records into flexible identity tokens. Although Luo et al.’s scheme has achieved relatively lower computation and gas consumption, our scheme offers an even broader application scenario. The premise of the scheme proposed by Yu et al. is that IIoT devices, device administrators, system management servers, edge servers, and other entities are completely trusted, which is a condition that is rarely met in real IoT scenarios. Wang et al.’s work novelly uses tokens to implement access control, but it suffers from high deployment and computational costs, as well as prolonged response times. In addition, our scheme undergoes rigorous performance testing, which further substantiates its reliability and effectiveness in the IoT context.

**Table 3 sensors-25-01857-t003:** Comparison of the proposed method with related works.

Related Work	Key Methods	Scenario	Attributes					
			**Dynamic** **Policy**	**Fine-Grained** **Control**	**Scalability**	**Automated** **Execution**	**Immutability**	**Distributed** **Storage**
Zhang et al. [23]	ABAC, Smart Contract	Smart City	✗	✓	✓	✓	✓	✗
Luo et al. [27]	ABAC, PDAC, Smart Contract	IIoT	✗	✓	✓	✓	✓	✗
Pu et al. [31]	SCR-BAC, Smart Contract	IoMT	✗	✗	✗	✓	✓	✗
Yu et al. [30]	Public and Private Key, Blockchain	IIoT	✗	✗	✗	✗	✓	✗
Lyu et al. [29]	MBAC, Blockchain, Token	ICN	✗	✗	✗	✓	✓	✗
Shi et al. [28]	ACl, Blockchain	Distributed IoT	✗	✓	✗	✗	✓	✗
Wang et al. [26]	Token, Smart Contract	IIoT	✗	✗	✗	✓	✓	✗
Sultana et al. [25]	Smart Contract	Smart Grid	✗	✓	✓	✓	✓	✗
Guo et al. [24]	DABAC, Smart Contract	IoMT	✗	✓	✗	✓	✓	✓
**Ours**	ABAC, RBAC, Smart Contract	IoMT	✓	✓	✓	✓	✓	✓

✓ Property held; ✗ Property missed.

In the scenario of IoMT, Pu et al. [31] discussed a risk- and SC-based access control model for medical data, which adjusted the doctor’s access rights by quantifying the risk value according to the doctor’s current and historical behavior. Meanwhile, Guo et al. [24] elucidated a domain attribute-based access control (DABAC). The solution realizes regional dynamic management and distributed deployment of devices by integrating smart gateways and smart contracts with domain elements. Compared with Pu et al.’s solution, simulation experimental results show that MedAccessX has lower gas and computing consumption under the premise of more fine-grained permission settings. Furthermore, MedAccessX provides more flexible access control than Guo et al.’s solution while reducing deployment and computing consumption.

Lyu et al. [29] combined a matching-based access control (MBAC) model and proposed a blockchain-based access token mechanism. It mainly addresses the problems of single-point failure and low efficiency of hierarchical access control that are common in many access control schemes. In addition, the scheme introduces Cuckoo filters to improve the query efficiency of tokens and designs a caching strategy to meet the universal caching of information-centric networking (ICN). In contrast, our access control scheme uses smart contracts combined with ABAC and RBAC models to achieve more fine-grained access control with lower computational consumption.

Shi et al. [28] utilized the hash address of the blockchain account as a user’s identity, defining their access control permissions and finally storing all relevant information in the access control list (ACL). This approach solves the problems of data tampering and leakage caused by the centralized access control mechanism used in traditional distributed IoT. In contrast, our MedAccessX uses encrypted userID for identity authentication, which further protects the privacy and security of users. Moreover, when it comes to access control, our parameter retrieval method is significantly faster than the approach taken by Shi et al. [28], offering a more efficient access management process.

Sultana et al. [25] focused on providing authenticated data sharing and authorized access control among IoT devices. Three smart contracts are used to achieve trust, authorization, and authentication in the IoT network. In contrast to MedAccessX, their method requires re-registering users every time their permissions are updated, resulting in an overly large list of stored policies. Our solution, on the other hand, focuses on the efficient matching of users with policies, effectively reducing deployment and computing consumption.

In comparison, MedAccessX seamlessly integrates both ABAC and RBAC. It not only realizes the basic properties of these access control frameworks but also has the flexibility to adopt different policy-making models tailored to various situations.

## 7. Conclusions and Future Work

We have introduced a dynamic access control framework that seamlessly integrates smart contracts with ABAC and RBAC models, aiming to address challenges in IoT data access control such as single point of failure, reliance on central authorities, limited scalability, and static permission management. Our solution, MedAccessX, employs four smart contracts on the blockchain to enable decentralized and flexible data access and sharing in IoT-based medical environments.

Through comprehensive evaluations, we analyzed MedAccessX’s deployment cost, throughput, average gas consumption, and response times for key functions. Additionally, we compared the performance of MedAccessX against existing access control schemes. These results highlight MedAccessX’s advantages in cost-effectiveness and processing efficiency, making it well-suited for IoMT applications.

**Future work:** While MedAccessX shows promising results, several limitations remain. First, the current framework does not yet support real-time data updates, which could enhance its responsiveness in dynamic IoMT environments. Future work could explore integrating real-time data synchronization to address this limitation. Second, although our framework supports ABAC and RBAC, the gas consumption of ABAC needs to be further optimized to improve the efficiency of the framework. Techniques such as policy caching could be investigated to reduce the overhead of ABAC evaluations. Third, although our framework can achieve data sharing by being deployed on a consortium blockchain, how to comply with relevant laws and regulations in the real world is also something that needs to be discussed in the future. Finally, we also aim to explore adaptive access control mechanisms that can automatically adjust policies based on user behavior patterns and contextual data, further enhancing the framework’s flexibility and adaptability.

## Figures and Tables

**Figure 1 sensors-25-01857-f001:**
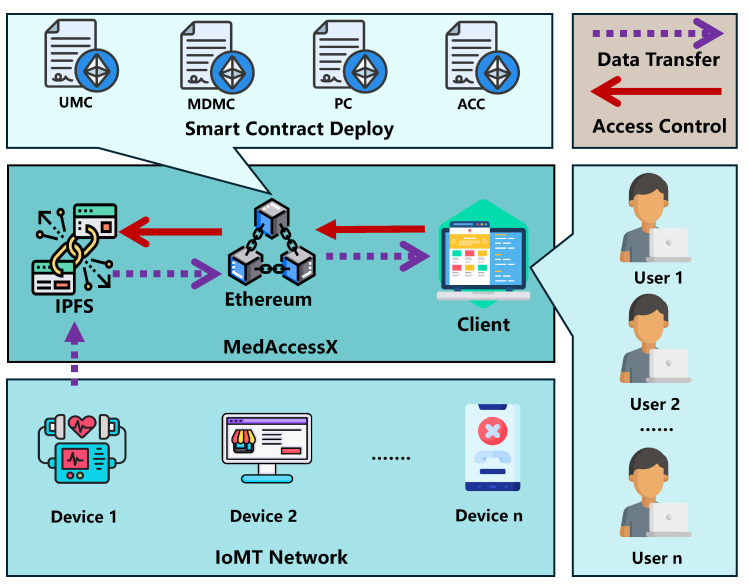
The framework of MedAccessX scheme.

**Figure 2 sensors-25-01857-f002:**
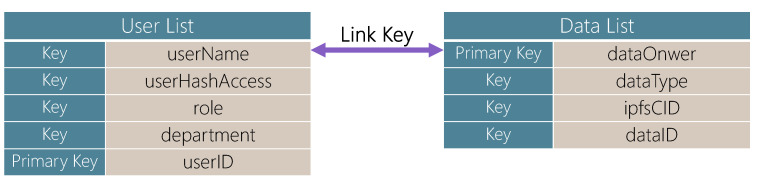
UserList and DataList.

**Figure 3 sensors-25-01857-f003:**
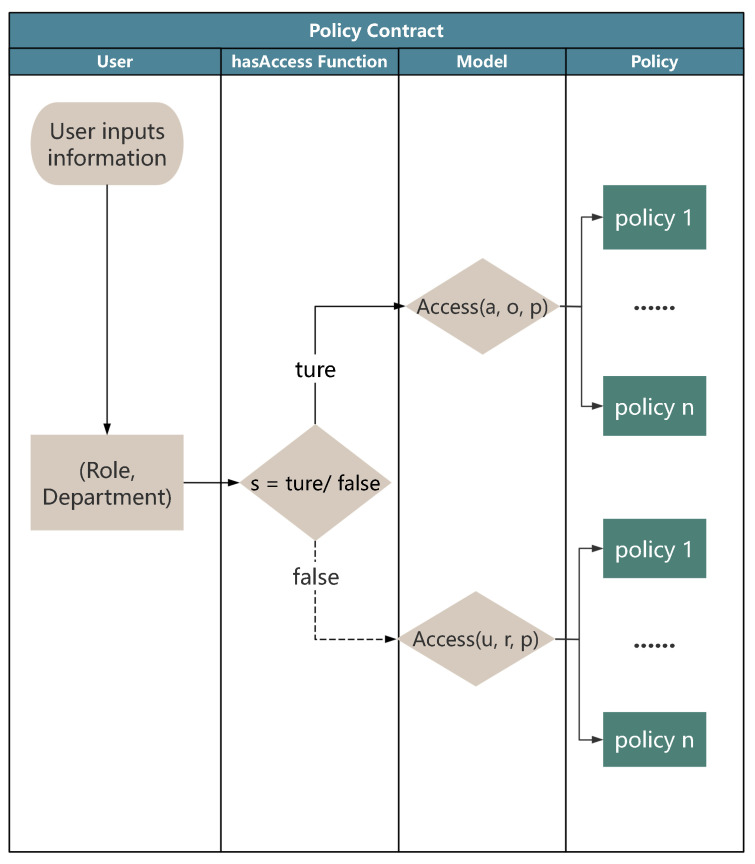
Policy selection process.

**Figure 4 sensors-25-01857-f004:**
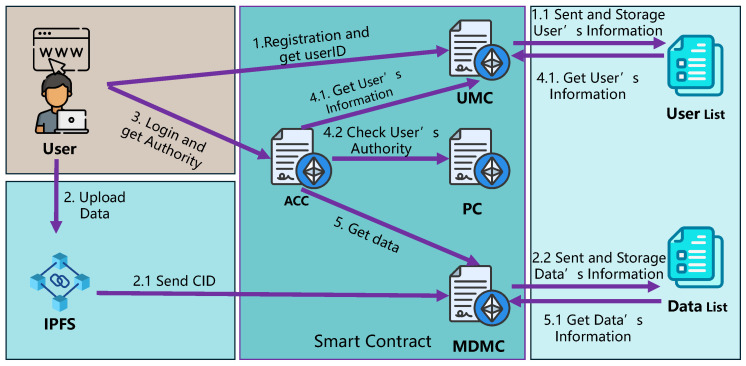
Workflow of AC process.

**Figure 5 sensors-25-01857-f005:**
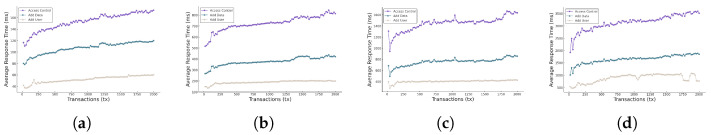
Average response time with different nodes. (**a**) ART with one node. (**b**) ART with five nodes. (**c**) ART with ten nodes. (**d**) ART with 20 nodes.

**Figure 6 sensors-25-01857-f006:**
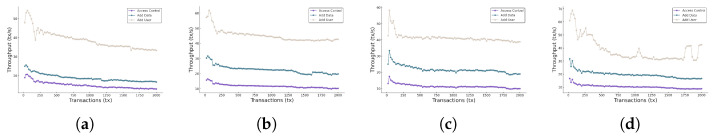
Throughput with different nodes. (**a**) TPS with one node. (**b**) TPS with five nodes. (**c**) TPS with ten nodes. (**d**) TPS with twenty nodes.

**Figure 7 sensors-25-01857-f007:**
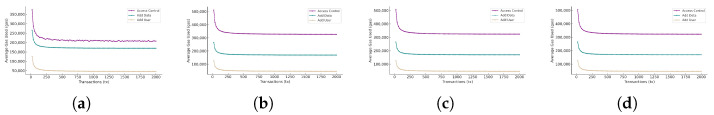
Average gas cost with different nodes. (**a**) AGC with one node. (**b**) AGC with five nodes. (**c**) AGC with ten nodes. (**d**) AGC with twenty nodes.

**Figure 8 sensors-25-01857-f008:**
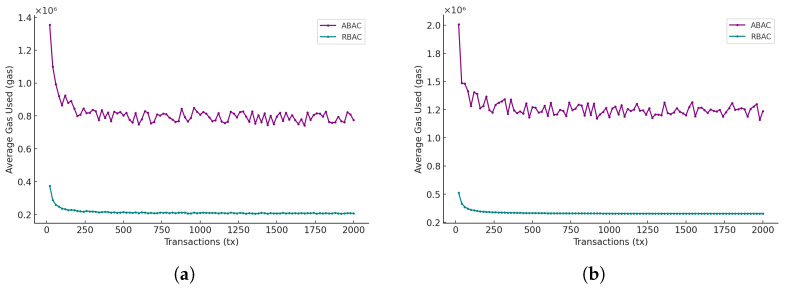
Average gas cost Comparison between ABAC and RBAC with different nodes. (**a**) Comparison with one node. (**b**) Comparison with five nodes.

**Figure 9 sensors-25-01857-f009:**
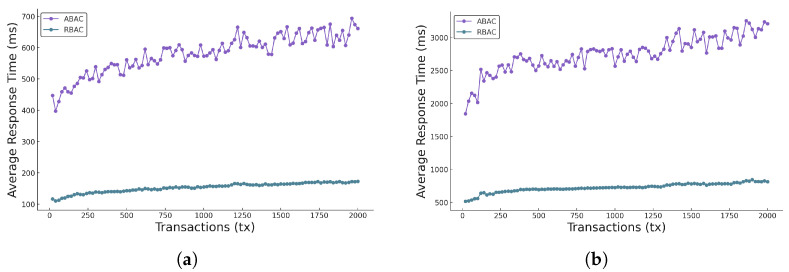
Average Response Time Comparison between ABAC and RBAC with Different Nodes. (**a**) Comparison with one node. (**b**) Comparison with five nodes.

**Figure 11 sensors-25-01857-f011:**
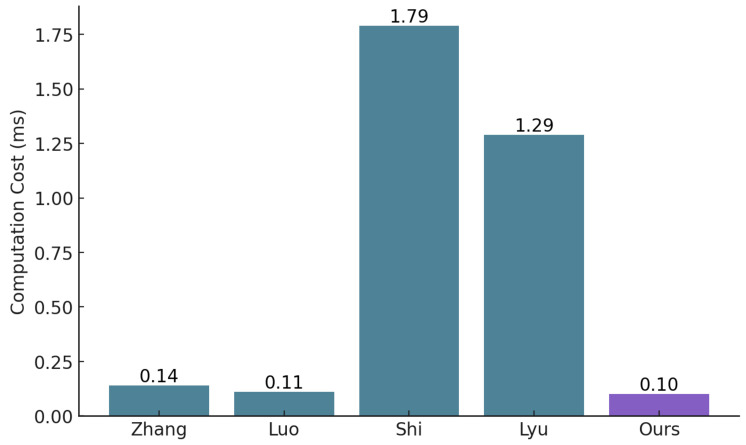
Comparison of the computational cost of the computational cost with Zhang et al. [23], Luo et al. [27], Shi et al. [28] and Lyu et al. [29].

**Table 1 sensors-25-01857-t001:** Subject and object of ABAC model.

**Subject**	**Var Type**	**Example**
userName	String	Alice
userAddress	Address	0x5B38Da6a701…f56beddC4
role	Enumeration Types	Chief Doctor
department	String	Cardology
userID	byte32	0x2a95f822717…9d51affed
**Object**		
dataOwner	String	John
dataType	Enumeration Types	MedicalRecord
ipfsCID	String	QmTzQ1Y9ft9i…w3TkRUxEgz
dataID	byte32	0xd4de688bfb…0b82bb848b

**Table 2 sensors-25-01857-t002:** Deployment cost of four contracts.

Smart Contract Type	Average Deployment Cost (Gwei)	In USD ($)
UMC	820,006	40.97
MDMC	775,320	38.74
PC	588,100	29.36
ACC	913,539	45.61
Total	3,096,965	154.65

## Data Availability

The original data presented in the study are openly available at https://github.com/Minfeng-Qi/medicalDataDapp.

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
