# Peer review of "MedAccessX: A Blockchain-Enabled Dynamic Access Control Framework for IoMT Networks"

_sensors, 2025, doi:10.3390/s25061857_

Round 1

Reviewer 1 Report

Comments and Suggestions for Authors

The paper presents a well-designed framework, MEDACCESSX, which integrates blockchain technology with attribute-based access control (ABAC) and role-based access control (RBAC) to address the challenges of data privacy, security, and access control in Internet of Medical Things (IoMT) networks. 

The following are the main contributions of the article:

1. The hybrid approach (ABAC and RBAC) allows for both dynamic and role-based access control, making the system highly adaptable to the complex and dynamic nature of IoMT environments.

2. The authors have conducted extensive experiments to evaluate the performance of MEDACCESSX, including deployment cost, gas consumption, throughput, and response time. 

3. The paper provides a detailed security analysis, addressing potential threats such as DoS attacks, Sybil attacks, and transaction order dependency attacks. 

The article can also be improved from the following three aspects:

1. Although the ABAC model provides flexibility, it incurs higher gas consumption and response times compared to RBAC. The authors could explore techniques such as policy caching or more efficient policy evaluation algorithms to reduce the overhead associated with ABAC.

2. The paper could make a discussion on the potential challenges of deploying MEDACCESSX in real-world healthcare environments, such as regulatory compliance (e.g., GDPR, HIPAA) and interoperability with existing healthcare IT systems.

3. The figures (e.g., Fig.1, Fig.4) are helpful, but some of them could be improved for better readability. For instance, Fig. 1 could include more descriptive labels or annotations to help readers understand the workflow more intuitively.

Author Response

Comments 1: Although the ABAC model provides flexibility, it incurs higher gas consumption and response times compared to RBAC. The authors could explore techniques such as policy caching or more efficient policy evaluation algorithms to reduce the overhead associated with ABAC.

Response 1: [Second, although our framework supports ABAC and RBAC, the gas consumption of ABAC needs to be further optimized to improve the efficiency of the framework.] Thank you for pointing this out. We agree with this comment. Therefore, we have clearly pointed out in the future work in the Conclusion that we will optimize the gas consumption of the ABAC model in the future to improve the efficiency of the entire framework. You can find this change in page 25, Conclusion section, 867-869 line.

Comment 2: The paper could make a discussion on the potential challenges of deploying MEDACCESSX in real-world healthcare environments, such as regulatory compliance (e.g., GDPR, HIPAA) and interoperability with existing healthcare IT systems.

Response 2: Agree. We have, accordingly, modified future work to emphasize this point. How to deploy our framework in the real world and comply with regulations is our follow-up work. You can find this change in page 25, Conclusion section, 870-873 line.

Comments 3: The figures (e.g., Fig.1, Fig.4) are helpful, but some of them could be improved for better readability. For instance, Fig. 1 could include more descriptive labels or annotations to help readers understand the workflow more intuitively.

Response 3: We agree with Comments 3. We have added more labels in Fig.1 and Fig.4 in page 7 and page 14.

Reviewer 2 Report

Comments and Suggestions for Authors

The main idea of the proposed blockchain-based data exchange system is to integrate ABAC and RBAC within smart contracts. The enabled dynamic access control supports privacy, security, and auditable data sharing among healthcare providers. The source code and experiment results can be accessed on GitHub. Overall, the quality of this work is good enough to be possible published in this journal. Several comments are listed as follows to help enhance the quality of the paper:

  1. The authors claimed the proposed data exchange system based on blockchain. The related processes are executed with smart contracts. However, the pseudocode and related figures seem independent from blockchain. The authors should highlight how the proposed system managed through smart contracts on the blockchain.
  2. The topic is “a blockchain-enabled dynamic access control framework”, but the cooperations between blockchain and access control are unclear.
  3. Some figures are unclear result in hard to reading.

Author Response

Comments 1:The authors claimed the proposed data exchange system based on blockchain. The related processes are executed with smart contracts. However, the pseudocode and related figures seem independent from blockchain. The authors should highlight how the proposed system managed through smart contracts on the blockchain.

Response 1: Thank you for pointing this out. We agree with this comment. Therefore, we have . You can find this change in page 8, 3.1 User Management Contract (UMC), 286-297 line; in page 9, 3.2 Medical Data Management Contract (MDMC), 324-331 line; in page 8, 3.1 User Management Contract (UMC), 286-297 line.

Comments 2:The topic is “a blockchain-enabled dynamic access control framework”, but the cooperations between blockchain and access control are unclear.

Response 2: Agree. We have, accordingly, modified future work to emphasize this point. We emphasize the connection of our framework to blockchain. You can find this change in page 6, Conclusion section, 224-227 line.

Comments 3:Some figures are unclear result in hard to reading.

Response 3: Agree. We have modified some of the unclear pictures and uploaded them in PDF format to ensure that the pictures will not be blurry after being enlarged.